REG γ knockdown suppresses proliferation by inducing apoptosis and cell cycle arrest in osteosarcoma

Yin Zhiqiang 1
Jin Hao 2
Huang Shibo 3
Qu Guofan guofanqu_doctor@163.com 4
Meng Qinggang mqg138456@163.COM 5
1 Department of Orthopedics, The Fourth Affiliated Hospital of Harbin Medical University , Harbin , China
2 Department of Orthopedics, The First Affiliated Hospital of Harbin Medical University , Harbin , China
3 Department of General Surgery, Heilongjiang Agricultural Reclamation General Hospital , Harbin , China
4 The Affiliated Tumor Hospital of Harbin Medical University, Harbin Medical University , Harbin , Heilong , China
5 Department of Medical Science Institute of Harbin, The Fourth Affiliated Hospital of Harbin Medical University, Harbin Medical University , Harbin , Heilong , China
Leppik Liudmila
Electronic publication date: 2020 Apr 15
Publication date: 2020
Volume: 8
Electronic Location ID: e8954
Received 2019 Nov 11; Accepted 2020 Mar 21
Copyright: ©2020 Yin et al.
Copyright year: 2020
Copyright holder: Yin et al.
License: This is an open access article distributed under the terms of the Creative Commons Attribution License, which permits unrestricted use, distribution, reproduction and adaptation in any medium and for any purpose provided that it is properly attributed. For attribution, the original author(s), title, publication source (PeerJ) and either DOI or URL of the article must be cited.
License URL: https://creativecommons.org/licenses/by/4.0/

Keywords: Osteosarcoma, Proliferation, REG γ

Funding: Natural Science Foundation of Heilongjiang Province, China QC2016102 H2016002 This study was supported by the Natural Science Foundation of Heilongjiang Province, China (Grant NO. QC2016102 and H2016002). The funders had no role in study design, data collection and analysis, decision to publish, or preparation of the manuscript.

==============================
Background

Osteosarcoma (OS) is the most common malignant bone tumor with high mortality in children and adolescents. REG γ is overexpressed and plays oncogenic roles in various types of human cancers. However, the expression and potential roles of REG γ in osteosarcoma are elusive. This study aims at exploring possible biological functions of REG γ in the pathogenesis of osteosarcoma and its underlying mechanism.

Methods

Quantitativereverse transcription-polymerase chain reaction (qRT-PCR), western blotting andimmunohistochemistry (IHC)were performed to detect the expression levels of REG γ in OS tissues and cell lines. Then, the effects of REG γ expression on OS cell proliferation in vitro were analyzed by Cell Counting Kit-8 (CCK-8), ethylene deoxyuridine (EdU), colony formation, flow cytometry. The protein levels of apoptosis and cell-cycle related proteins were evaluated using western blotting.

Results

In present study, we found for the first time that REG γ is overexpressed in osteosarcoma tissues and cell lines and knockdown of REG γ significantly inhibits cell proliferation and induces apoptosis and cell cycle arrest in osteosarcoma cells. Furthermore, we observed that p21, caspase-3 and cleaved caspase-3 are increased while the expression of cycinD1 and bcl-2 are decreased after REG γ depletion in osteosarcoma cells. In conclusion, REG γ may be involved in the proliferation of osteosarcoma and serve as a novel therapeutic target in patients with osteosarcoma.

Introduction

Osteosarcoma (OS) is the most common primary malignant bone cancer affecting children and adolescents (Botter, Neri & Fuchs, 2014). Although OS only accounts for less than 0.2% of all cancers, mortality rate of OS is up to 50% in children (Siegel, Miller & Jemal, 2016). Despite rapid development in treatment strategies, prognosis of patients with OS has shown no significant improvement in nearly 20 years (Kempf-Bielack et al., 2005; Marko, Diessner & Spector, 2016). As a result, there is an unmet need to identify novel molecules involved in the tumorigenesis of OS, which will be beneficial for treatment of patients with OS.

REG γ, also known as PSME3, PA28g or Ki antigen, is a member of the 11S family of proteasome activator and plays crucial roles in an ubiquitin- and ATP-independent non-lysosomal intracellular protein degradation (Ma et al., 1993). Previous researches showed that REG γ-knockout mice and cells display growth retardation, reduced cell proliferation and increased apoptosis (Li et al., 2013). Moreover, REG γ can promote the degradation of multiple tumor suppressor proteins (Chen et al., 2007; Li et al., 2015b; Li et al., 2006; Wang et al., 2015) and maintain stability of centrosome and chromosomal (Zannini et al., 2008). Additionally, growing evidence have confirm that REG γ is overexpressed in multiple types of cancers, such as skin cancer (Li et al., 2015a), breast cancer (Yi et al., 2017), renal cell cancer (Chen et al., 2018) and thyroid carcinoma (Zhang, Gan & Ren, 2012). However, the expression and biological functions of REG γ in OS have never been elucidated.

This study aimed to elucidate the expression and biological functions of REG γ in OS. We found that REG γ is overexpressed in OS tissues and cell lines compared with the adjacent normal tissues and the normal osteoblast hFOB1.19 cell, respectively. In addition, we found that knockdown of REG γ significantly inhibited proliferation and induced apoptosis and cell cycle arrest in MG-63 and SaoS-2 cell lines. Moreover, we also observed that multiple apoptosis and cell cycle related proteins expression were altered in REG γ-silenced osteosarcoma cells. Our results demonstrated that REG γ plays an oncogenic role in osteosarcoma and may be a molecular target in the treatment of patients with OS.

Materials and Methods

Clinical tissue samples

A total of ten OS tissues and ten adjacent normal tissues were obtained from primary osteosarcoma patients who underwent operative treatment at the Third Affiliated Hospital of Harbin Medical University from June 2018 to December 2018. Nine of the patients received preoperative chemotherapy. Partial tissue specimens were snap-frozen immediately in liquid nitrogen and stored at −80 °C until use. The other samples were fixed with 4% paraformaldehyde and then immunohistochemically stained. This study was approved by the Ethics Committees of the Third Affiliated Hospital of Harbin Medical University and written informed consent was obtained from each patient.

Cell culture

Human OS cell lines MG-63 and SaoS-2 were cultured in RPMI-1640 medium. The normal osteoblast hFOB1.19 cell was cultured in F-12 medium. All media were supplemented with 10% fetal bovine serum, 100U/ml penicillin, and 100 mg/ml streptomycin. Cells were cultured in a humidified incubator at 37 °C with 5% CO2. The two OS cell lines and hFOB1.19 were obtained from Cell Bank of the Chinese Academy of Sciences (Shanghai, China).

Immunohistochemistry

Tissue samples were fixed with 4% paraformaldehyde, dehydrated by a gradient series of ethanol, and then embedded in paraffin. The 4-µm sections were deparaffinized, rehydrated, and then stained with hematoxylin and eosin (H&E). Next, the tissue sections were subjected to antigen retrieval, be blocked with goat serum and incubated with a primary antibody at 4 °C overnight. Subsequently, the sections were incubated with a goat anti-rabbit secondary antibody for 20 min at room temperature and then for 30 min with Streptavidin-HRP peroxidase. Diaminobenzidine (DAB)-H2O2 was used as a substrate for the peroxidase enzyme. Then, the sections were stained with hematoxylin and dehydration. The primary antibody (REG γ rabbit polyclonal antibody) used for IHC analysis was purchased from Proteintech (catalog number: 14707-1-AP).

Transient transfection

Three small interfering RNAs specifically targeting human REG γ (siRNA-REG γ) and a nonspecific negative control oligo (siRNA-NC) were purchased from GenePharma (Shanghai, China). The sequences of siRNA-REG γ and siRNA-NC were shown in Table 1. Cell forward transfections were performed using Lipofectamine 2000 (Invitrogen; Thermo Fisher Scientific, Inc. USA) according to the manufacturer’s instructions. Briefly, 5 µl Lipofectamine 2000 and 5 µl siRNA (20 µM) were used in 6-well plate. Serum-free media (optiMEM) was used to dilute the siRNA and transfection reagents. Transfection efficiency of siRNA-REG γ-1 and siRNA-REG γ-2 was more than 50 percent, so they were used in following researches. Total RNA and total protein were extracted after 48 h and 72 h of transfection, respectively.

Table 1 Sequences of Si-RNA and primers.

Name:	Sequence:	
Si-negative control	Sense:5′-UUCUCCGAACGUGUCACGUTT-3′	
	Antisense: 5′-ACGUGACACGUUCGGAGAATT-3′	
Si-REG γ-1	Sense: 5′-GCAGAAGACUUGGUGGCAATT-3′	
	Antisense:5′-UUGCCACCAAGUCUUCUGCTT	
Si-REG γ-2	Sense: 5′-CCAAGGAACCAAGGUGUUUTT-3′	
	Antisense:5′-AAACACCUUGGUUCCUUGGTT-3′	
Si-REG γ-3	Sense: 5′-GGAUAGAAGAUGGAAACAATT-3′	
	Antisense: 5′-UUGUUUCCAUCUUCUAUCCTT-3′	
GAPDH:	Sense:5′-CCACTCCTCCACCTTTGAC -3′	
	Antisense: 5′-ACCCTGTTGCTGTAGCCA -3′	
REG γ	Sense: 5′-CTCCTGATACTGTAGCCTCTTGG -3′	
	Antisense: 5′-AGCATCTGGACCTCACACTTG -3′	

Western blot

Total protein was extracted from tissue samples or cultured cells by using a cold RIPA buffer (Beyotime Biotechnology, Shanghai, China) with protease inhibitor cocktail (Sigma-Aldrich, USA) on ice. Protein concentration was quantified by bicinchoninic acid (BCA) protein assay kit (Beyotime Biotechnology, Shanghai, China). Equal amount of protein was separated by 12% SDS-PAGE and transferred onto a polyvinylidene fluoride membrane (PVDF, Millipore, MA, USA (Catalog number: IPVH00010 and ISEQ00010)). Membranes were blocked with 5% fat-free milk in PBS for 2 h at room temperature and then incubated with a primary antibody at 4 °C overnight. After washed with TBST, membranes were incubated with an HRP-labeled secondary antibody for 1 h at 4 °C. Finally, membranes were washed three times, chemiluminescent reagent (BeyoECL Plus, Beyotime, Shanghai, China (Catalog number: P0018)) was added on the membranes and the specific signals were visualized by a Tanon Chemiluminescence Imaging System (Shanghai, China). The intensity was determined using ImageJ software. The expression level of every protein was independently detected. The primary antibodies used for WB were purchased from Proteintech (REG γ (catalog number:14907-1-AP), β- actin (catalog number:20536-1-AP)) and Cell Signaling Technology (p21 (catalog number:2947), bcl-2 (catalog number:2872), caspase-3 (catalog number:9662), cleaved caspase-3 (catalog number:9661), cyclin D1 (catalog number:2922)).

RNA isolation and qRT-PCR analysis

Total RNA was extracted from the OS tissue and cultured cells using TRIzol reagent (Invitrogen, CA, USA) and cDNA synthesis kit (Toyobo, Kyoto, Japan) was used in generating cDNA according to the manufacturer’s protocol. Quantitative reverse transcription-polymerase chain reaction (qRT-PCR) was performed using SYBR Green PCR kit (Toyobo, Kyoto, Japan) with an CFX96 Touch real time machine (Bio-Rad, USA) according to manufacturers’ instructions. DNA amplification was programmed for an initial 95 °C for 60 s, followed by 40 cycles of 95 °C 15 s, 60 °C 15 s and 72 °C 45 s, and ended with 72 °C 5min. The relative expression level of target gene mRNA normalized to β-actin was determined by the 2-ΔΔCt method. All the primers used in this study were showed in Table 1.

CCK-8 assay

Cells transfected with siRNA for 24 h were seeded in a 96-well plate (100 µl, 2,000 cells per well). The viability of transfected MG-63 and SaoS-2 cells were determined by a Cell Counting Kit (CCK-8, Dojindo, Tokyo, Japan) at 24, 48 and 72 h and 10 µl CCK-8 solution was added to each well before measurement. After 2 h of incubation, the optical density at 450 nm was detected by an ultraviolet spectrophotometer (Bio-Rad, Hercules, CA, USA).

Colony formation assay

After being transfected for 24 h, the cells were plated into 6-well plates at a density of 1.5 × 103/well and cultured for approximately five days until visible colonies formed. Cells were washed twice with cold PBS, fixed with 4% paraformaldehyde and then stained with 0.1% crystal violet (Beyotime Biotechnology, Shanghai, China). The number of colony formation was counted and photographed with a digital camera.

EdU assay

Cell proliferation was measured using the Cell-Light™ EdU Apollo®488 In Vitro. Imaging kit (Ribobio, Guangzhou, China (catalog number:C10310-1)) according to the manufacturer’s instructions. Briefly, MG-63 and SaoS-2 cells transfected with siRNA for 24 h were seeded into 96-well plates at a density of 6 ×103/well. After 24 h, cells were incubated with 50 µM EdU for 2 h. Then, the cells were fixed with 4% paraformaldehyde and the cell nuclei were stained with Hoechst 33342. Subsequently, the EdU-positive cells were captured and counted with a fluorescent microscope.

Apoptosis assay and cell cycle analysis

To analyze cells apoptosis rate, FITC-Annexin V Apoptosis Detection Kit (BD Biosciences, San Jose, CA (catalog number:556547)) was used according to the manufacturer’s instructions. After being transfected for 48 h, cultured cells were collected, washed twice with cold PBS and resuspended in 1 × binding buffer. Then, cells were stained with 5 µl Annexin V-FITC and 5 µl propidium iodide (PI) in the dark for 15 min at room temperature. For the cell cycle analysis, cells transfected with siRNA for 48 h were harvested, washed twice with precooled PBS, and fixed in 70% precooled ethanol at 4 °C overnight. Then, the cells were washed with precooled PBS and resuspended in 500 µl solution containing PI (50 µg/ ml) and RNase A (50 µg/ ml) in the dark at room temperature for 20 min. All the flow experiments were performed using BD FACS Calibur (Beckman Coulter, CA, USA). In addition, FlowJo_V10 software was used in cell apoptosis analysis and ModFit LT software was used in cell cycle analysis.

Statistical analysis

Data were presented as the mean ± SD and analyzed with Student’s t-test or one-way ANOVA by GraphPad Prism7.0. All tests were two-sided, and a P-value of < 0.05 was considered statistically significant. All experiments were independently performed three times.

Results

REG γ is upregulated in OS tissue and cell lines at both protein and mRNA levels

The expression of REG γ was examined in osteosarcoma tissues and adjacent normal tissues by using IHC, WB and qRT-PCR analysis. Clinical characteristics of included OS patients were shown in Table 2. Results demonstrated that REG γ expression was significantly overexpressed in OS tissues compared with adjacent normal tissues (Figs. 1A–1D). To further confirm the upregulated expression of REG γ in OS, western blot and RT-PCR analysis were performed in OS cell lines (MG-63 and SaoS-2) and human normal osteoblast (hFOB1.19) (Figs. 1E, 1F). Meanwhile, bioinformatic results from the Oncomine open cancer microarray database (https://www.oncomine.org/) also shown that REG γ mRNA levels were higher in sarcoma than in normal tissues (p <0.01) (Figs. 1G, 1H, 1I).

Table 2 Clinical characteristics of osteosarcoma patients.

	Age	Gender	Location	Size (cm)	Tumor stage	Metastasis	
Patient 1	15	F	Proximal
Fibula	3.5	IIA	No	
Patient 2	18	M	Proximal
Tibia	2.0	IB	No	
Patient 3	14	F	Distal
Femur	6.8	IIIA	Yes	
Patient 4	13	F	Distal
Femur	5.0	IIB	No	
Patient 5	14	M	Proximal
Tibia	5.5	IIIA	Yes	
Patient 6	26	M	Proximal
Tibia	2.5	IIA	No	
Patient 7	16	F	Distal
Femur	3.4	IIA	No	
Patient 8	23	M	Proximal
Tibia	4.2	IIB	No	
Patient 9	11	M	Proximal
Tibia	6.5	IIIB	Yes	
Patient 10	20	M	Distal
Femur	3.7	IIB	No	

Figure 1 REG γ expression isupregulated in OS.

(A–D) Expression of REG γ in OS tissues (T) and adjacent normal tissues (AT) as detected by IHC (A, B), WB (C) and qRT-PCR (D). In (B), the brown represents the expression of REG γ in OS tissues and AT. Pictures on the left and on the right are magnified 50 times and 100 times, respectively. (E, F) Expression of REG γ in two OS cell lines (MG-63 and SaoS-2) and a normal osteoblast cell line (hFOB1.19), as detected by WB (E) and qRT-PCR. (G, H, I). REG γ expression (median expression intensity) in sarcoma tissues and adjacent normal tissues derived from the Oncomine database (https://www.oncomine.org/). ∗∗P < 0.01, ∗P < 0.05.

SiRNAs targeting REG γ reduce the expression of REG γ at mRNA and protein level in OS cells

To reduce the expression of REG γ and avoid off-target phenomenon, the cells were transfected with three different siRNAs targeting REG γ and with Si-NC as control. The qRT-PCR analysis showed significantly decreased levels of REG γ mRNA in Si-REG γ-1 and Si- REG γ-2 groups compared to Si-NC group (p < 0.05) (Figs. 2A, 2B). Consistently, Si-REG γ-1 and Si- REG γ-2 also markedly inhibited the REG γ expression at protein levels as shown as in western blot analysis (Figs. 2C, 2D). Conclusively, Si-REG γ-1 and Si-REG γ-2 efficiently downregulated REG γ expression.

Figure 2 Si- REG γ reduce the expressionof REG γ.

Compared to Si-NC, Si- REG γ -1 and Si- REG γ -2 inhibit more than 50 percent of REG γ expression and Si- REG γ -3 inhibit less than 50 percent of REG γ expression at mRNA level (A, B) and protein level (C, D). Data are shown as the mean ± SD. ∗P < 0.05.

REG γ knockdown inhibits proliferation in MG-63 and SaoS-2 cells

To confirm REG γ biological functions in osteosarcoma, we performed a series of functional assays in cells after transfection. Compared to Si-NC, siRNA-REG γ-1 and siRNA-REG γ-2 were able to effectively suppressed OS cells growth determined by CCK-8 (p < 0.05) (Figs. 3A, 3B). Similarly, results of colony formation assay also demonstrated that the colon formation rates were obviously lower in REG γ silenced group than that in control group and gradually decreased in REG γ expression-dependent manner (Figs. 3C, 3D). In addition, data from EdU assay also revealed that REG γ depletion significantly decreased the number of cells in proliferative period (Figs. 3E, 3F). Taken together, we found that REG γ knockdown inhibits OS cells proliferation.

Figure 3 REG γ depletion suppresses OS cell progression in vitro.

REG γ depletion suppresses OS cell progression in vitro. (A, B) Effect of Si-REG- γ -1 and Si-REG- γ -2 on OS cell growth as determined by CCK-8 assay. (C, D) Representative OS cell colony formation images after transfection of Si-REG γ versus Si-NC. (E, F) Representative images of the EdU incorporation assay after transfection of Si-REG γ compared to after transfection of Si-NC in Mg-63 (E) and in SaoS-2 (F). Data are shown as the mean ± SD. ∗P < 0.05.

REG γ knockdown induces apoptosis and cell cycle arrest in OS cell lines

To elucidate the underlying mechanism of silencing REG γ inhibiting OS proliferation, we performed flow cytometry experiments and western blot analysis. We observed that the percentage of OS cells was increased in G0/G1 phase while decreased in S and G2/M phase following REG γ downregulation in flow cytometry (Figs. 4A–4D). Furthermore, we also measured the effect of REG γ knockdown on OS cells apoptosis and found that the reduction of REG γ dramatically increased OS cells apoptotic rate determined by flow cytometry (p < 0.05) (Figs. 4E–4H). In addition, we quantified the apoptosis and cell cycle related genes in OS cells after REG γ knockdown at protein levels. We found that the expression of bcl-2 was downregulated after transfected with siRNA-REG γ, while caspase-3 and cleaved caspase-3 were upregulated (Figs. 4I, 4J). Furthermore, cell cycle related genes were also significantly altered by siRNA- REG γ. Compared with Si-NC group, higher expression of p21 and lower expression of cyclinD1 were observed in siRNA-REG γ groups (Figs. 4C, 4D).

Figure 4 REG γ deficiency induce apoptosis and cell cycle arrest andalters multiple cell apoptosis and cell cycle related proteins in MG-63 andSaoS-2.

REG γ deficiency induce apoptosis and cell cycle arrest and alters multiple cell apoptosis and cell cycle related proteins in MG-63 and SaoS-2. Representative fl ow cytometry analysis of the cell cycle distribution of MG-63 and SaoS-2 cells transfected with Si- REG γ and Si-NC. (A–D) Apoptosis rate of MG-63 and SaoS-2 cells after transfection with Si-REG γ and Si-NC, as determined by fl ow cytometry (E–H). Si- REG γ alters apoptosis and cell cycle related genes at protein levels (I, J). Data are shown as the mean ± SD, ∗P < 0.05.

Discussion

In this study, we investigated the potential role of REG γ in human osteosarcoma. We revealed that REG γ was significantly overexpressed in human osteosarcoma tissues and cell lines for the first time. Knockdown of REG γ inhibited osteosarcoma cells proliferation, prompted cells apoptosis and cell cycle arrest, indicating that targeting REG γ could be an alternative for human osteosarcoma therapy in the future.

Murata et al. found that REG γ-deficient mice display growth retardation compared with wild type mice. Meanwhile, previous studies reported that REG γ was strongly implicated in many kinds of cancer cells proliferation (Chen et al., 2018; Li et al., 2015a; Zhang, Gan & Ren, 2012). Our results are consistent with these data, indicating that REG γ plays a critical role in osteosarcoma proliferation. To seek further the underlying mechanism that REG γ prompting human osteosarcoma proliferation, we used RNA interference to reduce REG γ expression in MG-63 and SaoS-2 human osteosarcoma cell lines, then performed cell-cycle profile and apoptosis analysis by flow cytometry and measured the cell cycle and apoptosis related proteins level.

Knockdown of REG γ lead more cells to be arrested at the G0/G1 phase in our experiments, that was parallel with other independent studies on prostate cancer cells (Chen et al., 2017) and renal carcinoma cells (Chen et al., 2018). However, another study also demonstrated that REG γ downregulation resulted in cell cycle arrest at the G2/M phase in HeLa cells (Chen et al., 2007). Therefore, we speculated that REG γ depletion lead to cell cycle arrest by different mechanisms in different cell types. Moreover, apoptotic percent of osteosarcoma cells transfected with Si-REG γ was higher than that in Si-NC group. Meanwhile, we found that proapoptotic protein (caspase-3) level was significantly increased, while antiapoptotic protein (bcl-2) was decreased following REG γ knockdown. Present results were supported by previous researches (Chen et al., 2018; Moncsek et al., 2015).

Additionally, we discovered that proteins regulating cell cycle also were altered by siRNA interference of REG γ. CyclinD1, known as a regulator of cyclin-dependent kinases, is indispensable in transition from G0/G1 phase to S phase and has been reported to serve as an oncogene in several types of cancers, including colorectal cancer (Wei et al., 2019), breast cancer (Hosseini et al., 2019) and osteosarcoma (Li et al., 2017). Therefore, we investigated the cyclinD1 expression and disclosed that level of cyclinD1 protein was gradually downregulated in Si-REG γ groups. Consistent with the phenomenon that more cells transfected with Si-REG γ were arrested at the G0/G1 phase than cells transfected with Si-NC. A previous study also has shown that the depletion of REG γ leads to a striking decrease in cyclinD1 levels in prostate cancer cells. Taking all these results into account, we demonstrated that cyclinD1 may play an important part in the REG γ-related control of cell cycle progression in osteosarcoma. In addition, p21 serves as a broad-spectrum cyclin-dependent kinases inhibitor and takes part in regulating the cell cycle in many types of cells (Weinberg & Denning, 2016). However, there has been significant debate as to whether the REG γ depletion increases p21 protein expression level. Previous studies revealed that the reduction of REG γ lead to a markedly increase in p21 in TPC cells, but only had a relatively slight effect on p21 levels in MCF-7 cells and even had no effect on p21 expression in HepG2 and 3T3-L1 cells (Chen et al., 2007), indicating a cell type-specific effect of REG γ depletion on p21 protein expression. Given distinct effect of REG γ depletion on different kinds of cells, we measured the p21 level in cells after transfection. Interestingly, we conformed for the first time that REG γ knockdown significantly increases the level of the p21 protein in osteosarcoma cell lines in the present study.

REG γ can regulate expression of multiple proteins, the specific mechanisms by which it plays its pivotal roles have been preliminarily elucidated in previous research. The cell cycle inhibitor p21 has also been identified to be a REG γ-proteasome target (Chen et al., 2007; Li et al., 2007). It was reported that the ubiquitin-independent REG γ-proteasome pathway is responsible for the degradation of p21 in vivo and in vitro. P53, tumor suppressor protein, have the ability to stimulate apoptosis and cell cycle arrest in the event of DNA damage and strongly suppress oncogenesis (Sharpless & DePinho, 2002). REG γ has been found to be a cofactor that assists the interaction between p53 and MDM2 by specifically binding to both proteins and promoting the ubiquitin-dependent proteasomal degradation of p53 (Zhang & Zhang, 2008). In addition, recent studies on post-translational modification of REG γ revealed an important regulatory mechanism for its activities. Researches have showed that REG γ can be phosphorylated in vitro and further identified the upstream kinase responsible for REG γ phosphorylation. It was found that MEKK3, a mitogen-activated protein kinase (MAPK) kinase, activates JNK and p38 MAPKs, directly binds with and phosphorylates REG γ (Hagemann, Patel & Blank, 2003). Another MAPK kinase, B-RAF, which activates ERK1/2 MAPK, has also been shown to interact with REG γ and phosphorylate REG γ in vitro (Hagemann, Patel & Blank, 2003; Hagemann & Rapp, 1999).

In this study, the underlying mechanisms of REG γ in promoting OS proliferation were not investigated in depth. Nevertheless, current study sheds light on a new perspective for OS treatment. Further researches assessing the deep contribution of REG γ in OS are warranted.

Conclusions

We revealed for the first that REG γ was significantly overexpressed in human osteosarcoma tissues and cell lines for the first time. Knockdown of REG γ inhibited osteosarcoma cells proliferation, prompted cells apoptosis and cell cycle arrest. We speculate that the REG γ-controlled proliferation of osteosarcoma cells is probably due to the role of REG γ in regulating the cell-cycle and apoptosis relevant proteins, but the specific mechanism is still unclear. Thus, further study is needed to comprehensively understand the importance of REG γ in the development and progression of osteosarcoma.

Supplemental Information

Supplemental Information 1 Raw data of Fig. 1

Click here for additional data file.

Supplemental Information 3 Raw data of Fig. 2

Click here for additional data file.

Supplemental Information 3 Raw data of Fig. 3

Click here for additional data file.

Supplemental Information 4 Raw data of Fig. 2

Click here for additional data file.

Supplemental Information 5 Raw data of Fig. 3

Click here for additional data file.

Supplemental Information 6 Raw data of Fig. 4

Click here for additional data file.

Supplemental Information 7 Raw data of Fig. 4

Click here for additional data file.

Additional Information and Declarations

Competing Interests

Author Contributions

Human Ethics

Data Availability

The authors declare there are no competing interests.

Zhiqiang Yin conceived and designed the experiments, performed the experiments, analyzed the data, prepared figures and/or tables, authored or reviewed drafts of the paper, and approved the final draft.

Hao Jin performed the experiments, authored or reviewed drafts of the paper, and approved the final draft.

Shibo Huang analyzed the data, prepared figures and/or tables, and approved the final draft.

Guofan Qu and Qinggang Meng conceived and designed the experiments, authored or reviewed drafts of the paper, and approved the final draft.

The following information was supplied relating to ethical approvals (i.e., approving body and any reference numbers):

This study was approved by the Ethics Committees of the Third Affiliated Hospital of Harbin Medical University and written informed consent was obtained from each patient.

The following information was supplied regarding data availability:

All the raw data are available in the Supplemental Files.

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
