# Peer review of "REG γ knockdown suppresses proliferation by inducing apoptosis and cell cycle arrest in osteosarcoma"

_PeerJ, doi:10.7717/peerj.8954_

## Round 0.1 · original submission · Major Revisions

Please provide the point by point response to the reviewers 3 and 4 comments. Pay special attention to the questions concerning Western blot analysis and provide proper description of the method and raw (unprocessed) photos of the blots.

Reviewer 1 ·

Basic reporting

This study focused on the effects of enforced REGgama expression on the proliferation of osteosarcoma. This paper is well organized and well written.

Experimental design

The experiments were properly designed, and the methods were suitable for the aims of this study.

Validity of the findings

This study provides some valuable information about the effects of REGgama overexpression in osteosarcoma. However, the roles of REGgama in cancer cells have been discussed in various kinds of tumors, and the related mechanism has been illustrated in other studies (as described in Instruction). Thus, to improve the novelty of this study, more study (not limited to cycle-cycle genes expression) are strongly suggested to explore the underlying mechanisms of REGgama.

Reviewer 2 ·

Basic reporting

The authors found for the first time that REG γ is overexpressed in osteosarcoma tissues and cell lines and knockdown of REG γ significantly inhibits cell proliferation and induces apoptosis and cell cycle arrest in osteosarcoma cells. Furthermore, they observed that p21 and caspase-3 are increased while the expression of cycinD1 and bcl-2 are decreased after REG γ depletion in osteosarcoma cells.

Experimental design

This article fails to meet the standards

Validity of the findings

The findings are far from the novelty. All the experimrnts are siRNA results.

Additional comments

The data quality are too low. The authors must use the stable knockdown cells for there study. In addition, xenograph model is reqiued.

·

Basic reporting

• Yin et al. had reported the role of REG γ in osteosarcoma patients and cell lines. The manuscript was written scientifically, but several concerns should be addressed before published.
• Line 25: Please specify the role of REGγ, like anti-apoptotic, or tumor suppressor protein.
• Line 31: What do you mean by “cell behavior”? Cell growth/ proliferation?
• Line 51: Please define those acronyms, like REG, PSME3, PA28g, etc.
• Line 53: Citation is needed for the statement on REGγ.
• Please include more findings like Zannini, et al. 2008 (DOI: 10.4161/cc.7.4.5355) on the role of REG γ in chromosomal stability (in introduction or discussion).
• Line 55: This statement is ambiguous. What cancer-related proteins are involved? Oncogenic proteins or tumor suppressor proteins? Please specify.
• Line 58: Should be “skin cancer”, not “skin carcinogenesis”.
• Line 71: The OS and adjacent normal tissues were employed to determine the expression of REG γ. However, I am concerning about the purity and integrity of these tissue samples. How the authors differentiate and confirm those tissues are “OS” or “normal” tissues? Is any histological examination like IHC on the mitotic antigen (Ki-67) or H&E staining performed to confirm the distribution/location of cancer cells? How do authors confirm there is no OS cancer cells had already infiltrated into the adjacent tissues?
• Line 73: According to the authors, seven OS patients received pre-operative chemotherapy and one is not. The authors may need to mention the main chemotherapeutic agents used as this may be the confounding factor for your apoptosis study. Is it possible that the pre-operative chemotherapy may somehow interfere with the result? Please explain this in the discussion section.
• Line 73: Besides, clinical features for the OS patients are missing. Information like age, gender, location of OS tumor, histological types of OS, grading and any metastasis should be reported.
• Line 143: The subheading “flow cytometry” is not suitable as the name of the assay will be a better fit. Please amend this into apoptosis assay and cell cycle analysis (or any similar).
• Figure 1A: Single IHC staining of REG γ is inconclusive to relate with OS cancer cells as authors did not identify the cancer cells and we are not sure the REGγ is expressed by which cells.
• Figure 1B: The authors mentioned there were 8 OA patient samples but figure 1B showed 10 results. Please explain this.
• Figure 1: Please standardize the acronym (AT & T) throughout the figures.
• Figure 1F: It would be better if the authors focus on osteosarcoma statistics. Describing the expression of REGγ in other sarcomas (leiomyosarcoma, liposarcoma, and fibrosarcoma) is not really contributing to this study. Suggest removing this data or convert this as supplementary data.
• Figure 2: How the authors calculate the relative REGγ mRNA level? Please include this under “Methodology”
• Figure 3A: The plot mark for siRNA-1 and -2 are the same.
• Figure 3A/B: The growth curve of si-NC is identical for both MG-63 and Saos-2 (same OD for 48 & 72h) which sounds weird to me as MG-63 is growing faster with shorter doubling time as compared to Saos-2 cells. Parallelly, the MG-63 cells could produce a larger/greater number of colonies due to its higher growth rate (Figure 3B). Please check again your data on Figure 3A.
• Figure 3C/D: Is it possible for authors to quantify the EdU levels by using ImageJ software?
• Figure 4A/B: Please manually set the maximum Y-axis scale (around 1600 in your study) to have a better comparison between groups. The default axis setting is “automatic” and this will make the G0/G1 curve for each group look similar. The G0/G1 peak should be higher for si-REG groups.
• Figure 4A/B: The asterisk marks are too small.
• Figure 4C/D: Apoptosis assessment on 48h-REGγ-knockdown cells revealed around 4% increase of apoptosis (for si-REG-2). Is 4% of apoptosis induction clinically relevant? Besides, upon knockdown of REGγ, there is only a 4% increase in apoptosis, is REGγ crucial for OS cell survival? I suggest authors amend the statement in line 40, 218 and 247.
• Figure 4E/F: The quantitative data for the protein levels is missing as authors claimed that the protein density had been quantified via ImageJ software (line 114-115). Authors may combine these two figures as 1, and repeated labeling is not necessary.
• Figure 4E/F: How multiple proteins are detected in this study? Stripping and reprobe method? Please describe this in methodology.
• Figure 4 title: The statement “alters multiple cell apoptosis and cell cycle-related genes in MG-63 and SaoS-2” not suitable as gene expression is not conducted. Suggest “apoptosis and cell cycle-related proteins in MG-63 and SaoS-2”.
• Line 195: The upregulation of caspase-3 (35kDa) could not directly explain/relate the apoptosis. This is because the 35kDa caspase-3 is an inactive zymogen (pro-caspase-3). The authors should report the cleaved subunit (p12 & p17) which are representing the activated caspase-3. Please amend the western blot and insert the blot with activated caspase-3 at 17 and/or 12 kDa position.
• Discussion: I am expecting more discussion from the authors. For instance, how REGγ regulates the CyclinD1, p21, and other proteins. Is it by transcriptional or post-transcriptional regulation? Since both MG-63 and Saos-2 cells are p53-mutated cell lines, authors may discuss the p53-independent p21 induction upon knockdown of REGγ. Is REGγ the upstream regulator or the intermediate signaling molecule? What is the transcriptional regulation of REGγ? Can it be upregulated by another oncogene like mutated p53 or kinases? What is the potential role of REGγ in OS carcinogenesis, progression and survival? What is the limitation of this study?
• Conclusion: An overall schematic diagram which concludes the findings is much appreciated.

Experimental design

• The experiment was conducted on OS cell lines and patients with siRNA approach, however, references and details are not adequately provided.
• Line 80: Any justification on why RPMI-1640 media were used for these OS cells as EMEM is recommended for MG-63, while McCoy’s 5A for Saos-2? Will this affect the REGγ expression and make comparisons across the studies become difficult?
• Line 88: Why authors need to use the H&E pre-stained tissues for subsequent IHC?
• Line 89: Please specify the name of the primary antibody used in the first place (combined from line 93-94), together with the catalog number for better reproducibility.
• Line 90: Is “overnight” in this study 12h?
• Line 91: Why streptavidin-HRP was used in this study? Is the secondary antibody linked with HRP?
• Line 99: Forward or reverse transfection? Please specify the amount/conc of siRNA and Lipofectamine reagent used? Is complete RPMI media (with FBS and antibiotics) or serum-free media like optiMEM was used to dilute the siRNA and reagents?
• Line 100: Is good to have the pictures to visualize the siRNA transfection efficiency as supplementary data. Is 50% transfection efficiency consider a good transfection?
• Line 109: Please include the PVDF catalog number.
• Line 113: Please include the chemiluminescent reagent used.
• Line 115-116: Please include the catalog number for these antibodies.
• Line 122: Please put in more details for the procedure like temperature and incubation time for each step to ensure good reproducibility.
• Line 125: What do you mean “Cell transfected with siRNA for 24h were seeded”? The authors trypsinized the cells after transfection for 24h? Why some experiment like apoptosis assay is using 48h-transfected cells?
• Line 140: Any justification on why 6000 cells/well were used in EdU assay but 2000 cells/wells was used for CCK-8 assay?
• Line 151: What is the PI concentration?
• Line 153: Please mention the use of ModFit LT software in the cell cycle analysis.
• Line 155: How about the normality of data?
• Line 170: Do authors confirm with the off-target phenomenon? Do authors determine the viability (trypan blue assay) and gene expression of the house-keeping gene after transfection? Cytotoxic effect of siRNA was shown to downregulate gene expression. How authors confirm that the si-REGγ-1 &-2 downregulate the REGγ gene expression is not via the cytotoxic effects of siRNA.

Validity of the findings

• Most of the findings sound legit and valid, but a second round of checking is required after this correction (refer to the comments in Basic Reporting).
• Please pay attention on the figure 3A (refer to the comments in Basic reporting).

Additional comments

N.A.

·

Basic reporting

The manuscript is well-constructed and well-written. The English used is good but the description in the results could be clearer and more concise.

Experimental design

The scope of the investigation is limited but clearly defined. The methods are well-described.

Validity of the findings

Some of the results are purely descriptive, like the western blot and colony formation. Description of the results should be more percise (statistical significance marked and stated in the text).

Additional comments

The authors investigated the role of REGg in the proliferation of osteosarcoma using SiRNA technique. The manuscript is well structured and well written. I only have minor comments as the following:
1. The visual display of western blot results is convincing. Did the author also quantify the band density?
2. Figure 1A: Can the authors point out in the legend and the figure the features to be considered?
3. Why was the relative mRNA expression in Figure 1C and 1E shown in different type of graphs?
4. Were the reduction of REGg expression caused by SiRNA significant? If yes, in Figure 2A and B, why were the significance not marked?
5. Were the colony formation quantified? The authors mentioned "obviously", do they mean "apparently"?
6. The authors should clearly indicate in the text that which results were significantly altered and which were not, maybe by indicating the p-value.
7. Some of the limitations should be clearly stated at the end of the discussion. For example, the mechanism of regulations of REGg in promoting cancer proliferation is not investigated in depth. Please mention also what are the purposed further studies and the current implications of current observation in cancer treatment.

Reviewer 5 ·

Basic reporting

1.The author should add the catalog number of antibodies and kits.
2.In fig4 E and F, the results were in same membrane? If yes, could the author describe the details of the methods of western? Because the MW of these proteins are similar, such as cyclinD and caspase3. Stripping buffer can be used, but the methods should be described. If no, the author should add more pic of loading control.

Experimental design

1.Author should add the protein level of Cl-caspase3 in fig 4 E and F.

Validity of the findings

No comment

---

## Round 0.2 · Minor Revisions

Dear Dr. Meng,

Thank you for significant improvements of your manuscript. Please address the minor comments from Reviewer 3 and also check the spelling (punctuation in Figure legends) in the text.

·

Basic reporting

Thank you authors for the 1st revision. I do have some follow-up questions for the previous comments:
-Comment no.9: Authors did not address my question directly. How the authors differentiate and confirm those tissues are “OS” or “normal” tissues based on H&E staining as H&E staining is not specific for cancerous cells. Additionally, Zhiqiang Yin and Hao Jin who performed the experiment, are affiliated from the Department of Orthopedics, not Oncology. Are they the right persons (equipped with proper training and qualifications) to detect and confirm the cancerous OS cells? Kindly advise.

Besides, as mentioned earlier, I am concerning about the purity and integrity of these tissue samples. According to the authors’ statement “Normal tissues were collected as far away from OS tissues as possible”, where in my opinion, it is not scientifically reproducible.

- Comment no.13: Authors claimed that H&E was performed before IHC. Are the same tissue sections were used in both H&E and IHC staining? I didn’t see any overlapping or side-by-side comparison of H&E and IHC in Figure 1.

- Comment no.18 & 19: I didn’t see any correction being done on Figure 3A/B in the amended version.

- Comment no. 20: Thank you authors for the effort to set the Y-axis. This is important to reduce the bias during comparisons between groups in cell cycle analysis. The G2/M peaks in Si- REG γ-2 groups were too low should not be a problem. Please amend the graphs by changing the Y-axis scale.

- Comment no.21: I didn’t see any correction being done on the asterisks of Figure 4A/B.

- Comment No.22: Authors did not address my question directly. I understand that the 4% of apoptosis induction is statistically significant but yet my question is whether this is clinically sound and relevant? Besides, upon knockdown of REG γ, there is only a 4% increase in apoptosis, is REGγ crucial for OS cell survival?

-Comment No.23: I am very appreciated the quantitative data of the protein levels in Figure 4. However, I am not sure how the authors calculated this. Is it by relative to the si-NC group? How many biological replicates that authors perform for Figure 4? Kindly describe in methodology.
Additionally, I suggest authors re-confirm the data. This is because I noticed that the quantitative number of protein bands for MG-63 caspase-3 si-REG-1 (1.90) is smaller than SaoS-2 p21 si-REG-1 (1.92), yet the band for MG-63 is much thicker. The si-NC for both is similar anyway.

- Comment No.24: I cannot find any reprobe description in line 113-114 method. Please correct this.

- Comment no.26: The anti-cleaved caspase-3 antibody (cat no.: 9661) from Cell Signaling Technology is not able to detect the cleaved subunit p12. It could detect p19 and p17 as demonstrated in Figure 4. Kindly change the 17kDa in Figure 4 to 19&17kDa as double bands were detected.

Experimental design

- Comment no.6: From my understanding, streptavidin-HRP is used to detect the biotin-labelled target. I am not sure how streptavidin-HRP peroxidase can be beneficial for quantity and distribution of REG γ in western blot. Kindly explain this.

- Comment no.9: PVDF membrane that used (catalogue no: IPVH00010) has the pore size of 0.45 µm (https://www.merckmillipore.com/MY/en/product/Immobilon-P-PVDF-Membrane,MM_NF-IPVH00010) which is not compatible with protein that smaller than 20kDa. In this case, the result for cleaved caspase-3 (17kDa) is affected where the protein may pass through the PVDF membrane and the blot result is not valid. Authors need to re-run the cleaved caspase-3 expression (17kDa) with 0.2 µm PVDF.

- Comment no.14: Sorry to say I am not sure what authors talking about. “Not so many cells were appreciated to leave cells enough living space”?? Are authors meaning that to use lesser cell conc in longer incubation CCK-8 assay?

Validity of the findings

No additional comment

Additional comments

No additional comment.

·

Basic reporting

No comment

Experimental design

No comment

Validity of the findings

No comment

Additional comments

Thank you for addressing my comments. I have no further comments about this manuscript.
I think the authors left out the "*" for statistical significance in Figure 2. This can be amended later.

Reviewer 5 ·

Basic reporting

The authors have revised comments.

Experimental design

The authors have added the protein level of the cl-caspase3.

Validity of the findings

No comment

---

## Round 0.3 · accepted · Accept

Thank you for significant improvements of your manuscript.